

**Influences of Entrainment-Mixing Parameterization on**
**Numerical Simulations of Cumulus and Stratocumulus Clouds**
Xiaoqi Xu[1,2,4], Chunsong Lu[1*], Yangang Liu[3*], Shi Luo[1,5], Xin Zhou[3], Satoshi Endo[3],
Lei Zhu[1], Yuan Wang[1]
1. Key Laboratory for Aerosol-Cloud-Precipitation of China Meteorological
Administration/Collaborative Innovation Center on Forecast and Evaluation of
Meteorological Disasters (CIC-FEMD), Nanjing University of Information Science &
Technology, Nanjing, China
2. Nanjing Joint Institute for Atmospheric Sciences, Nanjing, China
3. Environmental and Climate Sciences Department, Brookhaven National Laboratory,
Upton, US
4. Key Laboratory of Transportation Meteorology, CMA, Nanjing, China
5. College of Aviation Meteorology, Civil Aviation Flight University of China,
Guanghan, China
*    Correspondence: luchunsong110@gmail.com; lyg@bnl.gov
**Abstract**
Different entrainment-mixing processes can occur in clouds; however, a
homogeneous mixing mechanism is often implicitly assumed in most commonly used
microphysics schemes. Here, we first present a new entrainment-mixing
parameterization that uses the grid-mean relative humidity without requiring the



relative humidity of the entrained air. Second, the parameterization is implemented in
a microphysics scheme in a large eddy simulation model. Third, sensitivity experiments
are conducted to compare the new parameterization with the default homogeneous
entrainment-mixing parameterization. The results indicate that the new entrainment-
mixing parameterization has a larger impact on the number concentration, volume-
mean radius, and cloud optical depth in the stratocumulus case than in the cumulus case.
This is because inhomogeneous and homogeneous mixing mechanisms dominate in the
stratocumulus and cumulus cases, respectively, which is mainly due to the larger
turbulence dissipation rate in the cumulus case. Because stratocumulus clouds break up
during the dissipation stage to form cumulus clouds, the effects of this new entrainment-
mixing parameterization during the stratocumulus dissipation stage are between those
during the stratocumulus mature stage and the cumulus case. A large aerosol
concentration can enhance the effects of this new entrainment-mixing parameterization
by decreasing the cloud droplet size and evaporation time scale. This study sheds new
light on the improvement of entrainment-mixing parameterizations in models.
**1. Introduction**

The process of entrainment and subsequent mixing between clouds and their

environment is one of the most uncertain processes in cloud physics, which is thought
to be crucial to many outstanding issues, including warm-rain initiation and subsequent
precipitation characteristics, cloud-climate feedback, and evaluating the indirect effects





of aerosol (Paluch and Baumgardner, 1989; Yum, 1998; Ackerman et al., 2004; Kim et
al., 2008; Huang et al., 2008; Del Genio and Wu, 2010; Lu et al., 2011; Lu et al., 2014;
Kumar et al., 2013; Zheng and Rosenfeld, 2015; Fan et al., 2016; Gao et al., 2020; Gao
et al., 2021; Zhu et al., 2021; Xu et al., 2021; Kumar et al., 2013; Yang et al., 2016;
Yang et al., 2021). The most well-studied concepts are homogeneous/inhomogeneous
entrainment-mixing mechanisms. During homogeneous mixing, all droplets experience
evaporation, and no droplet is evaporated completely. During extremely
inhomogeneous mixing, some droplets near the entrained air evaporate completely,
while the remaining droplets maintain their original sizes. If the situation is somewhere
between these two extreme scenarios, an inhomogeneous mixing process occurs. Some
studies suggest that homogeneous mixing is likely to be typical (Jensen et al., 1985;
Burnet and Brenguier, 2007; Lehmann et al., 2009), whereas others have claimed that
extremely inhomogeneous scenario is dominant (Pawlowska et al., 2000; Burnet and
Brenguier, 2007; Haman et al., 2007; Freud et al., 2008; Freud et al., 2011). Different
mechanisms can be undistinguishable when the relative humidity in the entrained air is
high (Gerber et al., 2008).

Some sensitivity studies assuming homogeneous or extremely inhomogeneous

mixing have found that different mixing mechanisms can significantly influence the
microphysics and radiative properties of clouds (Lasher-Trapp et al., 2005; Grabowski,
2006; Chosson et al., 2007; Slawinska et al., 2008). For example, Grabowski (2006)
used a cloud-resolving model and found that the amount of solar energy reaching the



surface in the pristine case, assuming the homogeneous mixing scenario, is the same as
in the polluted case with extremely inhomogeneous mixing. This result was verified by
Slawinska et al. (2008) using a large-eddy simulation (LES) model. Although the
influence of different mixing mechanisms in simulations is lower when two-moment
microphysics schemes are used (Hill et al., 2009; Grabowski and Morrison, 2011;
Slawinska et al., 2012; Xu et al., 2020), Hill et al. (2009) also claimed that there are
still many uncertainties in the entrainment-mixing process, and the effect of different
mixing mechanisms can be more important over the entire cloud life-cycle.

In recent years, methods have been developed to describe general entrainment-

mixing processes, with homogeneous and extremely inhomogeneous scenarios as
special cases (Andrejczuk et al., 2006; Andrejczuk et al., 2009; Lehmann et al., 2009;
Lu et al., 2011). Hoffmann et al. (2019) and Hoffmann and Feingold (2019) conducted
LES at the subgrid-scale with turbulent mixing, using a linear eddy model. Andrejczuk
et al. (2009) used the results of direct numerical simulation (DNS) to establish a
relationship between instantaneous microphysical properties and Damköhler number
($D_a$, Burnet and Brenguier, 2007), and developed a parameterization of the entrainment-
mixing process. Lu et al. (2013) developed a parameterization of the entrainment-
mixing process based on the relationship between the homogeneous mixing degree ($\psi$)
and transition scale number ($N_L$) in the explicit mixing parcel model (EMPM), as well
as aircraft observation data. Gao et al. (2018) investigated how $\psi$ is related to $D_a$ and
$N_L$ in a DNS, to improve the parameterization of the entrainment-mixing process. Luo





et al. (2020) simulated more than 12,000 cases with EMPM by changing a variety of
parameters affecting entrainment-mixing processes and developed a parameterization
that improved the one proposed by Lu et al. (2013).
Although several entrainment-mixing parametrizations have been proposed, to the
best of our knowledge, only one study (Jarecka et al., 2013) has coupled an entrainment-
mixing parameterization with cloud microphysics to consider the change in cloud
droplet concentration during the entrainment-mixing process. Jarecka et al. (2013)
applied an entrainment-mixing parameterization, in terms of the Damköhler number, to
a two-moment microphysics scheme and found small impacts of entrainment-mixing
parameterization in shallow cumulus clouds. To further explore the influences of
entrainment-mixing processes, this study first modifies the entrainment-mixing
parameterization in terms of the transition scale number proposed by Luo et al. (2020)
to couple it more easily with microphysics schemes. The parameterization is then
implemented in the two-moment Thompson aerosol-aware scheme (Thompson and
Eidhammer, 2014). Finally, the effects of parameterization on the physical properties
of clouds are examined in both cumulus and stratocumulus clouds.
The rest of this paper is organized as follows: Section 2 describes the new
entrainment-mixing parameterization, simulated cases, and modelling setup. The major
results are presented and discussed in Section 3. The influences of the new entrainment-
mixing parameterization on cloud physics and the underlying mechanisms are
examined, and the effects of turbulence dissipation rate ($\varepsilon$) and aerosol concentration



are also discussed. Some concluding remarks are presented in Section 4.

**2. Parameterization, simulated cases, and modeling setup**
**2.1 The new entrainment-mixing parameterization**

According to Morrison and Grabowski (2008), the effect of the entrainment-

mixing process on cloud microphysical properties can be expressed as follows:
$$N_c = N_{c0} \left( \frac{q_c}{q_{c0}} \right)^{\alpha},$$
(1)

where $N_c$ and $N_{c0}$ are the cloud droplet number concentrations after and before the
evaporation process, respectively, and $q_c$ and $q_{c0}$ represent the corresponding cloud
water mixing ratios. It is noteworthy that when a new saturation is achieved after
evaporation, $q_c$ is determined by $q_{c0}$, relative humidity, air pressure, and temperature.
The parameter $\alpha$ can be pre-set to any value between 0 (homogeneous mixing) and 1
(extremely inhomogeneous mixing) to represent a different degree of subgrid-scale
mixing homogeneity. In this study, instead of specifying $\alpha$ as a predetermined constant,
here it is determined through expressions (Lu et al., 2013; Luo et al., 2020)
$$\alpha = 1 - \psi,$$
(2a)

$$\psi = c \exp(a N_L^b).$$
(2b)

where $a$, $b$ and $c$ are the three fitting parameters (Luo et al., 2020). The dimensionless
number $N_L$ is a dynamical measure of the degree of subgrid-scale mixing homogeneity
(Lu et al., 2011) defined by



$$N_L = \frac{L^*}{\eta},$$
(3a)

$$\eta = \left( v^3 / \varepsilon \right)^{1/4},$$
(3b)

$$L^* = \varepsilon^{1/2} \tau_{evap}^{3/2},$$
(3c)

where $L^*$ is the transition length (Lehmann et al., 2009), $\eta$ is the Kolmogorov microscale,
f$v$ is the kinematic viscosity; $\varepsilon$ is estimated following the method of Andrejczuk et al.

(2009):

$$\varepsilon = \left\langle \frac{1}{3} \left( u^2 + v^2 + w^2 \right) \right\rangle^{3/2} \Big/ L,$$
(4)

where $u$, $v$, and $w$ are the characteristic velocities in the horizontal and vertical
directions, respectively, and $L$ is the model grid size. The evaporation time scale ($\tau_{evap}$)
is defined as the time taken for droplets to evaporate completely in an unsaturated
environment, and is calculated as
$$\tau_{evap} = -\frac{r^2}{2 A S_e},$$
(5a)

$$A = \frac{1}{\left[ \left( \frac{L_h}{R_v T} - 1 \right) \frac{L_h \rho_L}{K T} + \frac{\rho_L R_v T}{D e_s(T)} \right]},$$
(5b)

where $r$ is the volume-mean radius of cloud droplets, $A$ is a function of pressure and
temperature, $S_e$ is the sub-saturation (relative humidity RH-1) of entrained air, $L_h$ is the
latent heat, $R_v$ is the specific gas constant for water vapour, $T$ is air temperature, $\rho_L$ is
the density of liquid water, $K$ is the coefficient of thermal conductivity of air, $D$ is the
diffusion coefficient of water vapour in the air, and $e_s(T)$ is the saturation vapour
pressure over a plane water surface at temperature $T$.



Unfortunately, $S_e$ in Equation (5a) is generally unavailable in atmospheric models,
including LES models. Thus, the entrainment-mixing parameterization developed by
Luo et al. (2020) based on the properties of entrained air cannot be used directly. To
solve this problem, we modify the entrainment-mixing parameterization of Luo et al.
(2020) by replacing $S_e$ with the domain-averaged relative humidity in the EMPM, after
entrainment but before evaporation, based on 12,218 cases:
$$\psi = 107.19 \exp(-1.99 N_L^{-0.29}). \tag{6}$$
Figure 1 shows the fitting results of the modified new entrainment-mixing
parameterization. Compared to the parametrization proposed by Luo et al. (2020), the
modified parameterization has similar $\psi$-$N_L$ distributions, but with a larger $N_L$ for the
same $\psi$, because the EMPM domain-averaged RH is larger than the entrained-air RH.
With this modification, $N_L$, $\psi$, and thus the effect of the entrainment-mixing processes
on droplet concentration can be directly calculated using the LES grid RH. It is
important to note that we do not assume that the RH of entrained air is equal to that of
the LES grid. Such a modification is only for the convenience of parameterization
applications in microphysics schemes. The details of the EMPM simulations and related
calculations are provided by Luo et al. (2020).

**2.2 LES model, simulation cases, and modelling setup**
The LES model is built by applying the large-scale forcing module presented in
Endo et al., 2015) to the Weather Research and Forecasting (WRF) model tailored for



solar irradiance forecasting (WRF-Solar, Hacker et al., 2016; Haupt et al., 2016). The
large-scale forcing data (VARANAL) used in this process is derived from the
constrained variational analysis (CVA) approach developed by Zhang et al. (2001) and
provided by the U.S. Department of Energy's Atmospheric Radiation Measurement
Program (www.arm.gov). The modified entrainment-mixing parameterization is
implemented in the two-moment Thompson aerosol-aware scheme (Thompson and
Eidhammer, 2014).

To investigate the behaviours of the new entrainment-mixing parameterization in

different cloud types, cumulus and stratocumulus cases are simulated. For both the
cumulus and stratocumulus cases, the horizontal resolution of the model is 100 m × 100
m with a domain area of 14.4 km × 14.4 km. The vertical direction is divided into 225
layers with a resolution of 30 m.

For each cloud case, $\psi$ is first set to 1 for the *default* experiment because most LES

models assume a homogeneous entrainment-mixing mechanism. The simulation with
the new entrainment-mixing parameterization (Equations (1-6)) is hereafter referred to
as *new*. First, $N_L$ is diagnosed for each grid, and $\psi$ is then calculated using Equation (6).
Finally, the variation in $N_c$ during entrainment-mixing is obtained using Equations (1)
and (2a).

Considering the significant impacts of $\varepsilon$ and initial droplet number concentration

on the entrainment-mixing process (Luo et al., 2020; Lu et al., 2013; Grabowski, 2006;
Hoffmann and Feingold, 2019), an alternative method that calculates $\varepsilon$ from the subgrid



turbulent kinetic energy (Deardorff, 1980) is also investigated and is referred to as
*new_tke*:
$$\varepsilon = CE^{3/2} / L, \qquad (7)$$

where $C = 0.70$ is an empirical constant and $E$ is the subgrid turbulent kinetic energy.
To examine the influence of the aerosol number concentration on the entrainment-
mixing process, we conduct the numerical experiments *default_10* and *new_10* by
multiplying the initial aerosol number concentrations, for the *default* and *new* models,
respectively, by a factor of 10. Thus, five sets of numerical experiments are conducted
for both the cumulus and stratocumulus cases; the names of the experiments and
corresponding descriptions are summarized in Table 1.

**3. Results**
**3.1 Cumulus case**
For the cumulus case, the simulation starts at 9:00 UTC on 11 June 2016 and ends
at 03:00 UTC on 12 June 2016 with an output interval of 10 min and spin-times of 3 h.
Figure 2 shows the temporal evolution of the cloud fraction from the five numerical
experiments. Grid points with $q_c$ larger than 0.01 g/kg are defined as "cloudy areas".
Also shown for comparison is observational data with a one-hour temporal resolution,
which is provided by the LES Atmospheric Radiation Measurement Symbiotic
Simulation and Observation (LASSO) campaign (Gustafson et al., 2020). The
observations show that the cloud forms at 12:00 UTC on 11 June and dissipates



completely by 01:00 UTC on 12 June with a maximum cloud fraction of 0.47 at 16:00
UTC on 11 June. All simulations capture the evolution of the cloud fraction and exhibit
similar values to the observational data.
Figure 3 shows the evolution of the microphysical and optical properties of clouds
in the cloudy areas of all simulation experiments, including $q_c$, $N_c$, droplet volume-
mean radius ($r_v$), cloud water path (CWP), and cloud optical depth ($\tau$). To visually and
simultaneously compare the change in cloud droplet concentration under different
aerosol concentrations, the maximum cloud droplet concentration ($N_{cmax}$) from *default*
is used to normalize $N_c$ in *default*, *new*, and *new_tke*, while $N_{cmax}$ from *new_10* is used
to normalize $N_c$ in *default_10* and *new_10*. The CWP is calculated as:

$$\text{CWP} = \int_0^H \rho_a q_c(z) dz, \tag{8}$$

where $\rho_a$ is the air density, $q_c(z)$ is the cloud water mixing ratio at each height ($z$), and
$H$ is the cloud thickness. The optical depth $\tau$ is estimated with

$$\tau = \frac{3}{2} \frac{1}{\rho_w} \int_0^H \frac{\rho_a q_c(z)}{r_e(z)} dz, \tag{9}$$

where $\rho_w$ is the water density and $r_e(z)$ is the effective radius of the cloud droplets at
each height ($z$). The time-averaged values of these physical properties of the clouds are
listed in Table 2 for convenience.
For the low aerosol number concentration, the simulations with the new
entrainment-mixing parameterization have smaller $N_c$ (34.91 cm$^{-3}$ and 35.53 cm$^{-3}$ for
*new* and *new_tke*, respectively) and larger $r_v$ (13.34 µm and 13.29 µm for *new* and
*new_tke*, respectively) than the default homogeneous simulation (35.78 cm$^{-3}$ for $N_c$ and





13.27 μm for $r_v$ in *default*). However, comparing *new* to *default*, the relative changes in
$N_c$ (–2.43%), $r_v$ (+0.53%), and $\tau$ (–0.99%) are small. The relative changes in *new_tke*
are even smaller. When the aerosol concentration increases ten-fold (*default_10* and
*new_10*), $q_c$, CWP, and $\tau$ increase according to the aerosol indirect effect (Peng et al.,
2002; Wang et al., 2019; Li et al., 2011; Wang et al., 2011). Meanwhile, $r_v$ decreases
significantly owing to the larger cloud number concentration. The effects of the new
entrainment-mixing parameterization also increase, for example, the change in $N_c$
increases from –2.43% (*new* compared to *default*) to –4.45% (*new_10* compared to
*default_10*), $r_v$ increases from +0.53% to +0.85%, and $\tau$ from –0.99% to –1.18%; the
reasons for these changes are discussed later. These small changes are similar to those
identified in previous cumulus studies (Jarecka et al., 2013; Hoffmann et al., 2019).

**3.2 Stratocumulus case**

The stratocumulus case is simulated from 9:00 UTC on 19 April 2009 to 03:00

UTC on 20 April 2009; the first three hours are set to be spin-up times. We examine the
stratocumulus region of the cloud base at ~2.1 km and the cloud top at ~2.3 km (cloud
thickness of ~200 m). The time series of the cloud fraction in the observed values and
five simulated datasets from 12:00 UTC to 24:00 UTC are shown in Figure 4. The
observed data show that the cloud fraction increases with time and peaks at 16:00 UTC.
All the simulations capture the main features of the cloud fraction. The simulated cloud
fraction has a value of 1 before 16:00 UTC, fluctuates from 16:00 UTC to 21:00 UTC,



and decreases sharply after 21:00 UTC. This period can be divided into three stages,
namely the mature stage, pre-dissipation stage, and dissipation stage.
As with the cumulus case, the temporal evolutions of the physical properties ($q_c$,
$N_c$, $r_v$, CWP, and $\tau$) of the clouds are shown in Figure 5. In contrast to the oscillating
changes exhibited by the physical quantities in the cumulus case (Figure 3), the physical
properties in the stratocumulus case exhibit a mostly smooth temporal evolution.
Furthermore, *default* and *new* exhibit clear distinctions during the early periods, but
these differences decrease during the dissipation stage. This is also the case with
*default_10* and *new_10*.
To compare the different behaviours of the simulation experiments at different
stages, the results at the mature and dissipation stages are analysed in detail. The mean
values of the main microphysical and optical properties of the clouds are summarised
in Table 3. As expected, the cloud microphysical and optical properties at the mature
stage are all larger than those at the dissipation stage. The effects of the new
entrainment-mixing parametrization are also more significant at the mature stage.
Compared to *default*, the *new* model results in a 7.27% smaller $N_c$, 2.42% larger $r_v$, and
5.77% smaller $\tau$ during the mature stage. During the dissipation stage, the changes in
$N_c$, $r_v$, and $\tau$ are –4.35%, +0.80%, and –2.56%, respectively. In contrast to the cumulus
case, *new_tke* is close to *new* and even has a slightly larger influence on cloud properties
than *new*, when compared to *default*, during both stages. The largest influence of the
new entrainment-mixing parametrization occurs during the mature stage when the





aerosol concentration is ten times greater. The differences in $N_c$, $r_v$, and $\tau$ between
*new_10* and *default_10* are –9.67%, +2.91%, and –5.39%, respectively, averaged over
the mature stage. The maximum differences in $N_c$, $r_v$, and $\tau$ are –10.71%, +6.37%, and
–7.72%, respectively. These differences are much larger than those reported by Hill et
al. (2009) who found that assuming extremely inhomogeneous mixing has a negligible
effect on stratocumulus simulations. Our results also prove the speculation of Hill et al.
(2009) that the mixing process might play an important role when the stratocumulus is
thin (~200 m in this study). Furthermore, implementing the new entrainment-mixing
parameterization has similar effects on cloud properties to those described by Hoffmann
and Feingold (2019) who used the linear eddy model to represent subgrid-scale
turbulent mixing. Note that stratocumulus clouds occur in most regions around the
world and are important contributors to the surface radiation budget (Wood, 2012;
Zheng et al., 2016; Wang et al., 2021; Wang and Feingold, 2009). Stratocumulus clouds
dominate in some regions and occur over 60% of the time as vast long-lived sheets,
such as the *semi-permanent subtropical marine stratocumulus sheets* (Wood, 2012). In
these regions, a 5.39%–5.77% decrease in $\tau$, caused by the new entrainment-mixing
parameterization is expected to have significant effects on the simulation of regional
radiative properties and climate change.

The averaged influences of the new entrainment-mixing parametrization over all

the simulation periods are also examined (Table 4). Quantitatively, the effect of the new
entrainment-mixing parameterization is much greater on stratocumulus clouds than on



cumulus clouds. Compared to *default*, *new* has an average change of –5.90% in $N_c$,
+1.49% in $r_v$, and –3.98% in $\tau$. When the aerosol concentration increases ten-fold, the
differences in $N_c$, $r_v$, and $\tau$ between *default_10* and *new_10* are −8.97%, +2.77%, and
−3.56%, respectively. These differences are larger than the largest changes in the
cumulus case.

**3.3 Mechanisms of the effects of the new entrainment-mixing parameterization**
The different effects of the new entrainment-mixing parameterization on different
types of clouds and even on different stages of stratocumulus clouds are likely be related
to variations in the dominant mixing mechanism. To confirm this, we calculate the
average $\psi$ at all grid points experiencing evaporation, the proportion of inhomogeneous
mixing grid points to all grid points experiencing evaporation, and the average $\psi$ at the
inhomogeneous mixing grid points in *new*, *new_tke*, and *new_10* (Table 5) for the
cumulus case, and mature and dissipation stages in the stratocumulus case.
For the cumulus case, all three simulations exhibit large $\psi$ and a small proportion
of inhomogeneous mixing, indicating that homogeneous mixing is the dominant
entrainment-mixing mechanism in all three simulations (Luo et al., 2020; Lu et al.,
2013), especially in *new_tke*. Correspondingly, the influences of the new entrainment-
mixing parameterization on the cloud physical properties are not significant, as shown
in Figure 3 and Table 2. The *new_10* model exhibit a smaller average $\psi$ and a larger
proportion of inhomogeneous mixing than *new* and *new_tke*, which results in larger





changes in cloud physics, as mentioned in Section 3.1.

For the stratocumulus case, Table 5 shows the average $\psi$ at all grid points

experiencing evaporation, the proportion of inhomogeneous mixing grid points to all
grid points experiencing evaporation, and the average $\psi$ at the inhomogeneous mixing
grid points during the two stages. The mature stage always has a smaller $\psi$ but a larger
proportion of inhomogeneous mixing than the dissipation stage. The inhomogeneous
mixing process dominates the mature stage in *new* and *new_tke*, because more than 60%
of the grid points experience inhomogeneous mixing. The inhomogeneous mixing
process is more dominant in *new_10*, because less than 3% of the cloudy grid points
experience a homogeneous mixing process during the mature stage, which explains
why *new_10* has the largest influence when implementing the new entrainment-mixing
parametrization. Meanwhile, the average $\psi$ in both stages is smaller than that in the
cumulus case for the same simulation configuration. Thus, the effects of the new
entrainment-mixing parameterization are more significant for stratocumulus than for
cumulus clouds, especially at the mature stage. It is noted that the average $\psi$ and the
proportion of inhomogeneous mixing at the dissipation stage of *new* and *new_tke* in the
stratocumulus case are very close to the results of *new_10* in the cumulus case. This is
because the cloud fraction decreases sharply during the dissipation stage; the
stratocumulus clouds break up and produces cumulus clouds with small cloud droplet
radius.



**3.4 The effects of dissipation rate and aerosol concentration on the entrainment-**
**mixing process**
Previous studies have shown the notable effects of the dissipation rate and aerosol
concentration on the entrainment-mixing process. For example, Luo et al. (2020)
changed $\varepsilon$ from $10^{-5}$ m$^2$ s$^{-3}$ to $10^{-2}$ m$^2$ s$^{-3}$ and noted huge differences in the corresponding
$\psi$. Small et al. (2013) compared aircraft observations with different background
concentrations and found that higher pollution flights tended to slightly more
inhomogeneous mixing; Jarecka et al. (2013) also showed various homogeneities of
subgrid mixing when aerosol concentration increases ten-fold. To explain the different
behaviours of different simulations with the new entrainment-mixing parameterization,
the influences of $\varepsilon$ and aerosol concentration are examined. Figure 6 shows the
probability distribution functions (PDFs) of $\varepsilon$, $r_v$, $\tau_{evap}$, and $N_L$ for cloud grids
experiencing entrainment-mixing processes in *new*, *new_tke*, and *new_10* for the
cumulus and stratocumulus cases, respectively. The PDFs from the mature and
dissipation stages of the stratocumulus case are shown in Figure 7.

**3.4.1 Dissipation rate**
According to Equation (3), $N_L$ is a function of $\varepsilon^{3/4}$; hence, the PDF of $\varepsilon$ directly
affects $N_L$ and further results in different $\psi$. For the cumulus case, the mean $\varepsilon$ (0.0032
m$^2$ s$^{-3}$) in *new* is close to the value of 0.0043 m$^2$ s$^{-3}$ in *new_tke*; these values are similar
to those obtained for cumulus clouds in previous studies (e.g. Lu et al., 2016; Hoffmann



355 et al., 2019). The different $\varepsilon$ distributions cause a significant difference in the proportion

356 of inhomogeneous mixing (Table 5). As shown in Figure 1, cloud grids experience a

357 homogeneous mixing process if $N_L$ is larger than ~$10^5$, the limited distribution of $N_L$

358 values less than $10^5$ in *new_tke* results in a very small number of cloud grid points

359 undergoing inhomogeneous mixing process. Even at the cloud grid points that undergo

360 inhomogeneous mixing, the average $\psi$ is large (98.62%), because most of the $N_L$ values

361 are larger than $10^3$. Therefore, the cloud properties in *new_tke* are close to those in

362 *default*. The *new* model contains more frequent occurrences of small $\varepsilon$ and $N_L$ values,

363 resulting in more cloud grid points undergoing a more inhomogeneous mixing process

364 and exhibiting a smaller $\psi$, compared to those in the *new_tke* model (Table 5); however,

365 the results of *new* are still close to those of *default*, because of the dominance of $N_L$

366 values larger than $10^5$.

367   For the stratocumulus case, the mean values of $\varepsilon$ ($2.7\times10^{-4}$ m$^2$ s$^{-3}$ in *new* and

368 $2.9\times10^{-4}$ m$^2$ s$^{-3}$ in *new_tke*) are an order of magnitude less than those in the cumulus

369 case. Therefore, compared with the cumulus case, the distribution of $N_L$ is reduced in

370 the stratocumulus case, while the peak values of *new* and *new_tke* are similar and almost

371 reach the criterion of inhomogeneous mixing (~$10^5$). For the two stages of

372 stratocumulus clouds, $\varepsilon$ is an order of magnitude smaller, but $r_v$ was larger (Figure 7)

373 during the mature stage than during the dissipation stage. According to Equation (5a),

374 droplets with smaller $r_v$ are more prone to complete evaporation and have a smaller $\tau_{evap}$.

375 The combination of smaller $\varepsilon$ and larger $r_v$ results in a smaller $N_L$ (Equation (3)). This





is the reason for the new entrainment-mixing parametrization having more significant
effects during the mature stage than during the dissipation stage. In addition, the
similarity of the $\varepsilon$ and $r_v$ values during the dissipation stage of the stratocumulus case
in *new*, compared to the cumulus case in *new_10* (Figures 6a and 6b), explains the
similar average $\psi$ values of these scenarios and the proportion of inhomogeneous
mixing (Table 5).

Therefore, the distribution of $\varepsilon$ has a vital impact on the influence of the new

entrainment-mixing parameterization. Smaller values of $\varepsilon$ result in the new
entrainment-mixing parameterization having a more significant influence. Moreover,
the $r_v$ in the stratocumulus case is smaller than that in the cumulus case, which is also
conducive to a more inhomogeneous mixing process. These are the reasons why the
implementation of the new entrainment-mixing parameterization has a larger influence
in the stratocumulus case than in the cumulus case, when compared to a homogeneous
mixing mechanism.

**3.4.2 Aerosol concentration**

The aerosol concentration affects the entrainment-mixing process by decreasing

the cloud droplet radius. As $r_v$ decreases, the distributions of $\tau_{evap}$ in *new_10* moves to
a smaller overall value, while the mean value is an order of magnitude smaller than that
in *new*, which causes a much smaller $N_L$ because $N_L$ is proportional to $\tau_{evap}^{3/2}$ (Equations
(3a) and (3b)). The larger percentage of smaller $N_L$ values indicates that in *new_10*,



more grid points undergo an inhomogeneous mixing process, and the proportion of such
grid points is much larger than in the *new* model (Table 5). Therefore, compared to *new*,
*new_10* exhibit a smaller $\psi$ and the effects of the new entrainment-mixing
parameterization on cloud properties are more obvious, for both the cumulus and
stratocumulus cases.

**4. Concluding remarks**
The entrainment-mixing process near cloud edges has important effects on cloud
microphysics, but the most commonly used microphysics schemes simply assume one
extreme mechanism, that is, homogeneous entrainment-mixing. This study first
improves the entrainment-mixing parameterization proposed by Luo et al. (2020),
which connects the homogeneous mixing degree and transition scale number to
estimate the homogeneity of the subgrid mixing process and its impact on the droplet
number concentration. The improved parameterization uses grid-mean relative
humidity and can be implemented directly into microphysics schemes; there is no need
to know the relative humidity of the entrained air. Second, the modified entrainment-
mixing parameterization is implemented in the two-moment Thompson aerosol-aware
scheme of the LES version of WRF-Solar, to examine its effects on the microphysical
and optical properties of cumulus and stratocumulus clouds. Third, several sensitivity
experiments are conducted to investigate the effects of the new entrainment-mixing
parameterization under different conditions of turbulence dissipation rate and aerosol





number concentration.

Unlike the commonly assumed homogeneous mixing scenario, the new

entrainment-mixing parameterization produces a smaller cloud droplet number
concentration and larger cloud droplet radius, with the degree of difference depending
on cloud types and stages. Sensitivity tests show that in the cumulus case, the largest
average influence of the new entrainment-mixing parameterization occurs under a high
aerosol background, but results in only a 4.45% decrease in cloud droplet number
concentration and a 0.85% increase in cloud droplet volume-mean radius. The changes
become even smaller with a low aerosol background because of the larger cloud droplet
radius. In contrast, the new entrainment-mixing parameterization has a larger influence
on the microphysical and optical properties of stratocumulus clouds, especially under a
high aerosol background and during the mature stage, with a cloud fraction equal to 1.
The largest changes resulting from the new entrainment-mixing parameterization are
−9.67%, +2.91%, and −5.39%, for cloud number concentration, cloud droplet volume-
mean radius, and cloud optical depth, respectively. The new entrainment-mixing
parameterization has less of an influence on the dissipation stage than on the mature
stage of the stratocumulus case, but affects this case more than the cumulus case.

The varying effects of the new entrainment-mixing parameterization are caused

by variations in the dominant entrainment-mixing mechanism between different cloud
types and stages. Compared to the cumulus case, the stratocumulus case has a much
smaller homogeneous mixing degree and a larger proportion of inhomogeneous mixing





grid points, especially during the mature stage, which indicates that the inhomogeneous
mixing mechanism dominates in the stratocumulus case, while the homogeneous
mixing mechanism dominates in the cumulus case. As mentioned above, the changes
in physical properties of stratocumulus clouds in the dissipation stage are between those
in the mature stage and those of the cumulus case; this is because stratocumulus clouds
dissipate sharply to form small cumulus clouds, and the degree of homogeneous mixing
during the dissipation stage is therefore between that which occurs during the mature
stage and the cumulus case.
Sensitivity studies show that how turbulence dissipation rate and aerosol
concentration are treated in a simulation can have notable effects on the subgrid
homogeneity of the mixing process. A larger dissipation rate can accelerate the mixing
process, which results in a larger transition scale number and homogeneous mixing
degree; and therefore a mostly homogenous mixing mechanism. This is why the
cumulus case exhibit smaller changes than the stratocumulus case after the new
entrainment-mixing parameterization is implemented. Larger aerosol number
concentrations cause a smaller cloud droplet radius. Smaller droplets evaporate more
easily, which leads to a smaller transition scale number and a smaller homogeneous
mixing degree. Thus, the entrainment-mixing mechanism tends to be inhomogeneous.
Therefore, a larger aerosol number concentration increases the influence of the new
entrainment-mixing parameterization in both the cumulus and stratocumulus cases.
Note that the new entrainment-mixing parameterization could be more important



in the LES model if the relative humidity near the cloud is more accurately simulated,
because numerical diffusion may humidify the entrained air (Hoffmann and Feingold,
2019). The artificially increased relative humidity limits the influences of the new
entrainment-mixing parameterization, because homogeneous and inhomogeneous
entrainment-mixing processes are close to each other under conditions of high relative
humidity.

**Author contributions.** XX, CL and YL designed the experiments. XX carried out the
experiments and conducted the data analysis with contributions from all coauthors. XX,
CL, XZ, and SE developed the model code. XX prepared the paper with help from CL,
YL, YW, SL, and LZ.

**Competing interests.** The authors declare that they have no conflict of interest.

**Acknowledgements.** This research is supported by the National Key Research and
Development Program of China (2017YFA0604001), the National Natural Science
Foundation of China (41822504, 42175099, 42027804, 41975181, 42075073). Liu is
supported by the U.S. Department of Energy Office of Science Biological and
Environmental Research as part of the Atmospheric Systems Research (ASR) Program.
Brookhaven National Laboratory is operated by Battelle for the U.S. Department of
Energy under Contract DE-SC00112704. The large-scale forcing data used in this paper



can be downloaded from the U.S. Department of Energy's Atmospheric Radiation
Measurement Program with https://adc.arm.gov/discovery/#/results. The LASSO data
can be downloaded from https://archive.arm.gov/lassobrowser.

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



Table 1. Summary of names and corresponding descriptions of the five experiments for
each case of cumulus and stratocumulus. The meaning of each symbol for each
experiment can be found in the text.

| | Entrainment-mixing parameterization | Dissipation rate | Aerosol number concentration |
|---|---|---|---|
| *default* | $\alpha = 0$ | - | default |
| *new* | $\alpha = 1 - \psi,$ $\psi = 107.19\exp(-1.99N_L^{-0.29}).$ | $\varepsilon = \left\langle \frac{1}{3}\left(u^2 + v^2 + w^2\right)\right\rangle^{3/2} \Big/ L.$ | default |
| *new_tke* | $\alpha = 1 - \psi,$ $\psi = 107.19\exp(-1.99N_L^{-0.29}).$ | $\varepsilon = CE^{3/2} / L.$ | default |
| *default_10* | $\alpha = 0$ | - | default×10 |
| *new_10* | $\alpha = 1 - \psi,$ $\psi = 107.19\exp(-1.99N_L^{-0.29}).$ | $\varepsilon = \left\langle \frac{1}{3}\left(u^2 + v^2 + w^2\right)\right\rangle^{3/2} \Big/ L.$ | default×10 |






682 Table 2. Summary of the case mean values of the key quantities in all the simulations

683 of the cumulus case, containing cloud water mixing ratio ($q_c$), cloud droplet number

684 concentration ($N_c$), cloud droplet volume-mean radius ($r_v$), cloud water path (CWP),

685 and cloud optical depth ($\tau$). The experiments are detailed in Table 1.

|  | *default* | *new* | *new_tke* | *default_10* | *new_10* |
|---|---|---|---|---|---|
| $q_c$(g/kg) | 0.44 | 0.44 | 0.44 | 0.56 | 0.57 |
| $N_c$(cm$^{-3}$) | 35.78 | 34.91 | 35.53 | 278.80 | 266.37 |
| $r_v$(μm) | 13.27 | 13.34 | 13.29 | 7.05 | 7.11 |
| CWP(g/m$^2$) | 142.30 | 143.15 | 144.25 | 186.52 | 187.82 |
| $\tau$ | 13.07 | 12.94 | 13.02 | 31.29 | 30.92 |







Table 3. Summary of the case mean values of the key quantities in all the simulations
of the stratocumulus case, including cloud water mixing ratio ($q_c$), cloud droplet number
concentration ($N_c$), cloud droplet volume-mean radius ($r_v$), cloud water path (CWP),
and cloud optical depth ($\tau$). The numbers in and out of the parentheses are the results at
the mature and dissipation stages, respectively. The experiments are detailed in Table 1.

| | *default* | *new* | *new_tke* | *default_10* | *new_10* |
|---|---|---|---|---|---|
| $q_c$(g/kg) | 0.13 | 0.13 | 0.13 | 0.16 | 0.16 |
| | (0.039) | (0.039) | (0.039) | (0.041) | (0.041) |
| $N_c$(cm$^{-3}$) | 35.74 | 33.14 | 33.11 | 256.82 | 231.98 |
| | (19.76) | (18.90) | (18.82) | (138.74) | (126.99) |
| $r_v$(μm) | 10.32 | 10.57 | 10.65 | 5.15 | 5.30 |
| | (7.53) | (7.59) | (7.69) | (4.02) | (4.11) |
| CWP(g/m$^2$) | 41.39 | 41.33 | 41.78 | 56.21 | 57.12 |
| | (2.57) | (2.45) | (2.43) | (2.71) | (2.77) |
| $\tau$ | 4.68 | 4.41 | 4.40 | 13.17 | 12.46 |
| | (0.39) | (0.38) | (0.38) | (0.78) | (0.78) |




Table 4. Cloud water mixing ratio ($q_c$), cloud droplet number concentration ($N_c$), cloud
droplet volume-mean radius ($r_v$), cloud water path (CWP), cloud optical depth ($\tau$) in all
simulations for the entire lifetime of the stratocumulus case. The experiments are
detailed in Table 1.

|  | *default* | *new* | *new_tke* | *default_10* | *new_10* |
|---|---|---|---|---|---|
| $q_c$(g/kg) | 0.11 | 0.11 | 0.11 | 0.13 | 0.13 |
| $N_c$(cm$^{-3}$) | 29.98 | 28.21 | 28.12 | 223.65 | 203.59 |
| $r_v$(μm) | 9.38 | 9.52 | 9.57 | 5.06 | 5.20 |
| CWP(g/m$^2$) | 30.78 | 30.34 | 29.92 | 42.22 | 43.19 |
| $\tau$ | 4.02 | 3.86 | 3.89 | 10.39 | 10.02 |





Table 5. Homogeneous mixing degree ($\psi$) at all grid points experiencing evaporation,
the proportion of inhomogeneous mixing grid points to all grid points experiencing
evaporation, and $\psi$ at the inhomogeneous mixing grid points in the experiments *new*,
*new_tke* and *new_10* (Table 1) for the cumulus (Cu) and stratocumulus (St) case. The
numbers in and out of the parentheses are the results at the mature and dissipation stages
in the stratocumulus (St) case, respectively. The experiments are detailed in Table 1.

| | $\psi$ at all grids (%) | Proportion of inhomogeneous mixing grids (%) | $\psi$ at the inhomogeneous mixing grids (%) |
|---|---|---|---|
| *new* (Cu) | 98.80 | 13.78 | 91.21 |
| *new_tke* (Cu) | 99.93 | 4.52 | 98.62 |
| *new_10* (Cu) | 94.41 | 49.32 | 86.01 |
| *new* (St) | 79.38 | 60.22 | 71.33 |
| | (94.12) | (42.36) | (90.66) |
| *new_tke* (St) | 78.56 | 63.07 | 71.56 |
| | (94.68) | (40.61) | (89.33) |
| *new_10* (St) | 68.23 | 97.11 | 65.18 |
| | (88.22) | (73.19) | (85.12) |






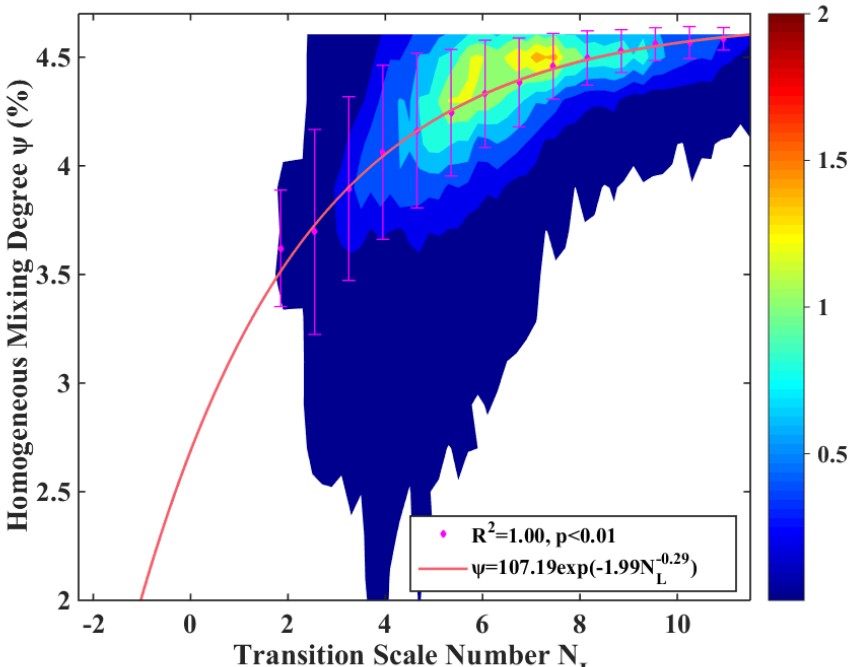


Figure 1. Parameterization of cloud entrainment-mixing mechanisms by relating
homogeneous mixing degree ($\psi$) to transition scale number ($N_L$) from EMPM. The
contours represent the joint probability distribution function (PDF) of $\psi$ vs $N_L$. The
magenta dots and error bars are mean values and standard deviations of $\psi$ in each $N_L$
bin, respectively. The mean values are fitted using a weighted least squares method with
the number of data points in each $N_L$ bin as the weight. The fitting equation, coefficient
of determination ($R^2$), and $p$-value are also given. $N_L$ is calculated by with the domain-
averaged relative humidity after entrainment but before evaporation in the EMPM.



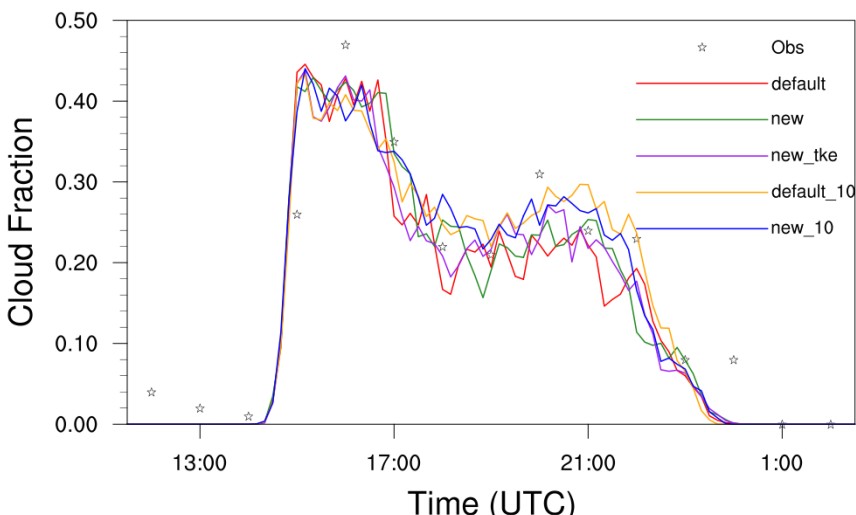


Figure 2. Time series of cloud fraction from 12:00 UTC on June 11, 2016, to 03:00

UTC on June 12, 2016, from the observation in LASSO and five simulation

experiments in the cumulus case. The five experiments are detailed in Table 1.




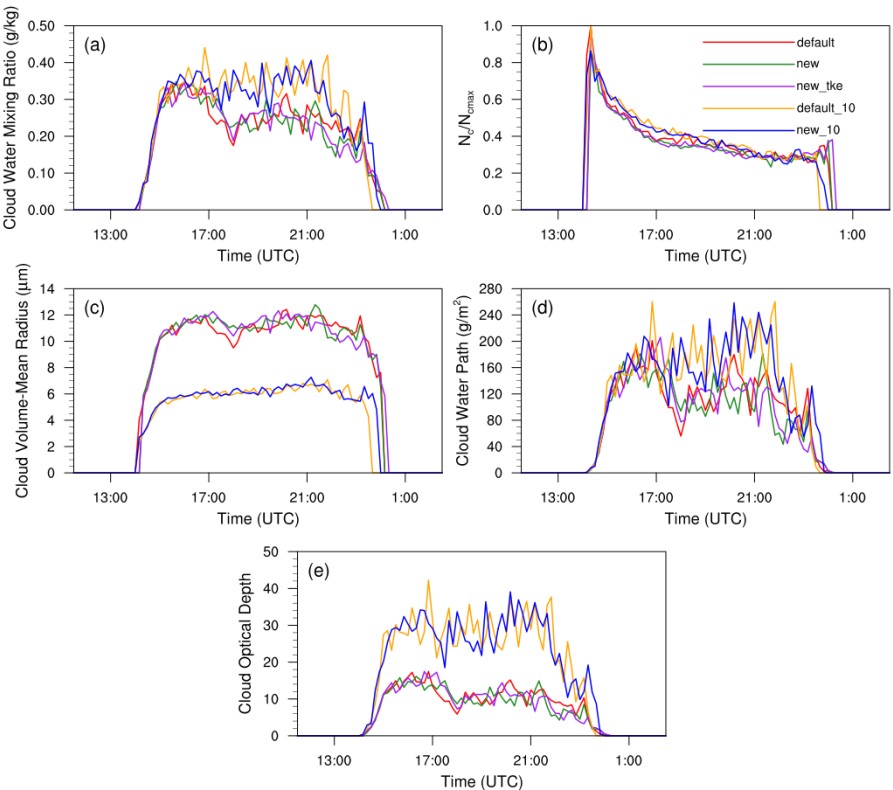


Figure 3. The temporal evolutions of main cloud microphysical and optical properties

in all simulation experiments for the cumulus case, including (a) cloud water mixing

ratio ($q_c$) (g/kg), (b) cloud droplet number concentration ($N_c$) (/cm$^3$), (c) cloud droplet

volume-mean radius ($r_v$) (μm), (d) cloud water path (CWP) (g/m$^2$), and (e) cloud optical

depth ($\tau$). In (b), $N_c$ in the experiments *default*, *new*, and *new_tke* are normalized by the

maximum cloud droplet concentration ($N_{cmax}$) from *default*, respectively; $N_c$ in the

experiments *default_10* and *new_10* are normalized by $N_{cmax}$ from *default_10*,

respectively. The five experiments are detailed in Table 1.

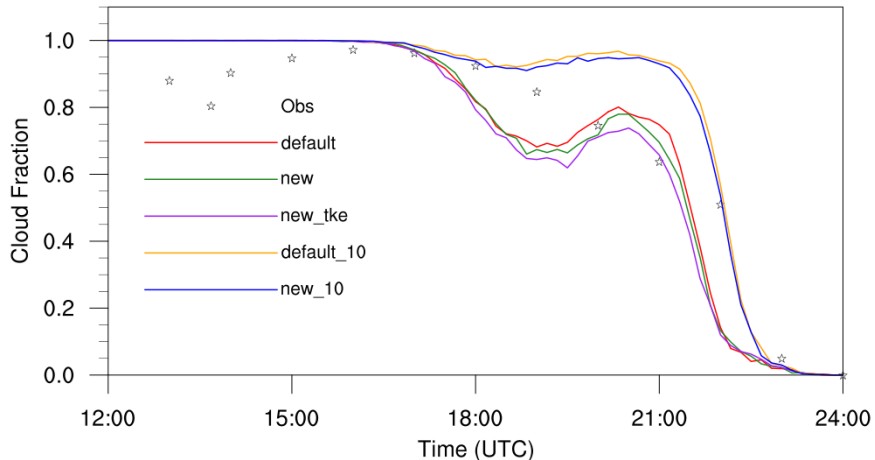


Figure 4. Time series of cloud fraction from the observation and five simulations in the
stratocumulus case. The five experiments are detailed in Table 1.

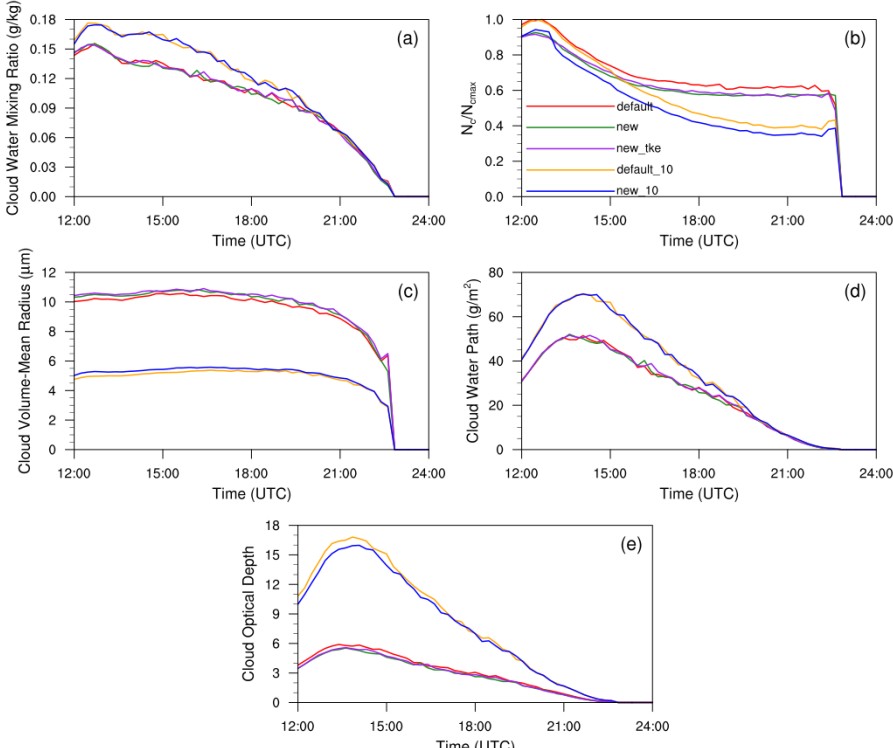


Figure 5. The temporal evolutions of main cloud microphysical and optical properties
in all simulation experiments for the stratocumulus case, including (a) cloud water
mixing ratio ($q_c$) (g/kg), (b) cloud droplet number concentration ($N_c$) (/cm$^3$), (c) cloud
droplet volume-mean radius ($r_v$) (μm), (d) cloud water path (CWP) (g/m$^2$), and (e) cloud
optical depth ($\tau$). In (b), $N_c$ in the experiments *default*, *new*, and *new_tke* are normalized
by the maximum cloud droplet number concentration ($N_{cmax}$) from *default*, respectively;
$N_c$ in the experiments *default_10* and *new_10* are normalized by $N_{cmax}$ from *default_10*,
respectively. The five experiments are detailed in Table 1.





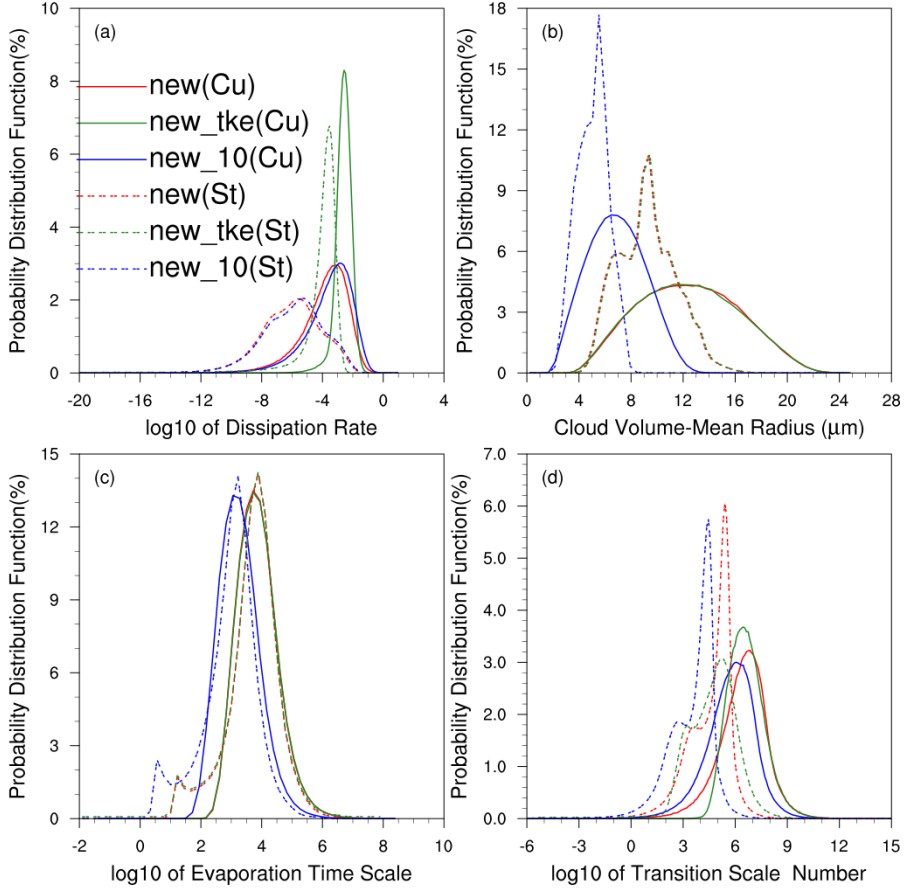


Figure 6. Probability distribution functions (PDFs) of (a) turbulence dissipation rate ($\varepsilon$),

(b) cloud droplet volume-mean radius ($r_v$), (c) evaporation time scale ($\tau_{evap}$), and (d)

transition scale number ($N_L$) of cloud grids experiencing the entrainment-mixing

process in the simulations with the new entrainment-mixing parameterization for the

cumulus case (Cu, the solid lines) and the stratocumulus case (St, the dash lines),

respectively. The experiments are detailed in Table 1.



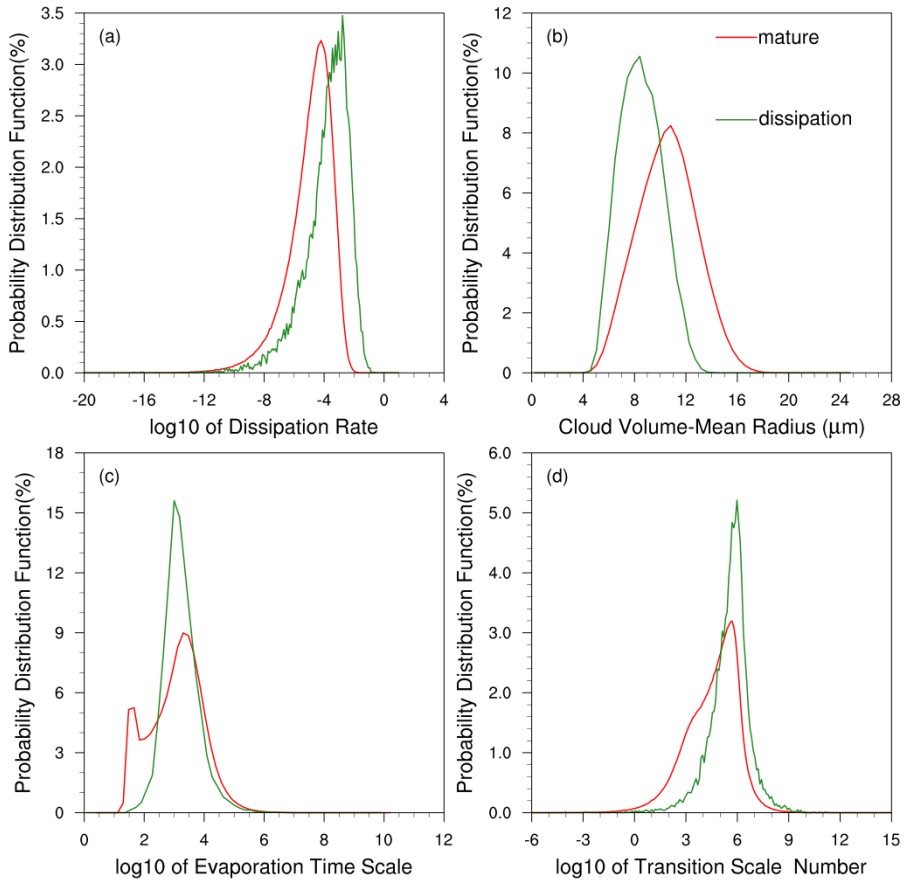

749

Figure 7. Probability distribution functions (PDFs) of (a) turbulence dissipation rate ($\varepsilon$),

(b) cloud droplet volume-mean radius ($r_v$), (c) evaporation time scale ($\tau_{evap}$), and (d)

transition scale number ($N_L$) of cloud grids experiencing the entrainment-mixing

process at the mature stage from 12:00 UTC to 16:00 UTC (the red lines) and the

dissipation stage from 21:00 UTC to 24:00 UTC (the green lines) in *new* for the

stratocumulus case. The experiment is detailed in Table 1.