# Peer review of "Influences of Entrainment-Mixing Parameterization on Numerical Simulations of Cumulus and Stratocumulus Clouds"

_Atmospheric Chemistry and Physics, 2021_

## Referee Comment (RC1)

**Review**

Title: **Influences of Entrainment-Mixing [1] Parameterization on Numerical Simulations of Cumulus and Stratocumulus Clouds**

Authors: Xiaoqi Xu Chunsong , Yangang Liu, Shi Luo, Xin Zhou, Satoshi Endo, Lei Zhu, Yuan Wang

**Summary**

The authors provided a new method for entrainment-mixing parametrization in the LES model. This scheme uses the grid mean RH and can be applied directly in microphysics schemes in the models. They have tested their method in LES version of WRF-Solar simulation for cumulus and stratocumulus clouds. Also, they have conducted the experiments for the sensitivity analysis for different turbulent dissipation rate and aerosol number concentrations.

This study provides a good method for parametrization. However, before publication in ACP, the authors have to clarify few questions provided below.

**Comments.**

Major:

The new scheme is based on the parameters $\alpha$ and the $\psi$. However, $\psi$ depends on $S_e$ (sub saturation of entrained air) which was taken from another model EMPM. My main question is how this new approximated value of Se is validated?  It seems that you are taking a parameter from one model and improving another model.  It should be validated by some observation or any other reliable source.

In addition, the model results presented in this study should be validated using observation data or a well-established theory.

---

## Referee Comment (RC2)

**Review of "Influences of Entrainment-Mixing Parameterization on Numerical Simulations of Cumulus and Stratocumulus Clouds" by Xu et al. (ACP-2021-937)**

The character of entrainment-mixing can have substantial impacts on the microphysical composition of clouds, with commensurate effects on the cloud optical properties and hence their role in the climate system. Nonetheless, most cloud models neglect the natural variability in the mixing process and assume homogeneous mixing as the default. For the application in bulk cloud models, this study develops a parameterization to consider the natural range between homogeneous and extreme inhomogeneous mixing based on simulation data derived from a high-resolution stochastic turbulence model applied in a previous study of the authors (Luo et al. 2020). After development, the parameterization is applied to a cumulus and a stratocumulus case and the results are analyzed.

The manuscript is generally well-written, interesting, and of relevance. However, there might be a fundamental flaw in the assumptions made in the parameterization development that can invalidate the entire manuscript, as I will outline further below. Therefore, I cannot recommend the publication of the manuscript at this stage.

**Major Comments**

*Developing an entrainment-mixing parameterization that depends on the grid-averaged relative humidity.* While I understand that it is necessary to simplify physical processes for parameterization purposes, the presented parameterization glances over crucial factors in the entrainment-mixing process. The entrainment-mixing process depends heavily on the relative humidity RH of the entrained air. While I agree that this variable is not directly accessible in most models, using the grid-averaged RH can be misleading if the fraction of entrained air $f$ is unknown. To illustrate this, we make the simplifying assumption that RH mixes linearly. (More rigorous calculations can be based on, e.g., Paluch (1979).) With this simplification, we find that

$$RH_{entrained} = \frac{RH_{grid} - (1-f)RH_{cloud}}{f},$$

where the subscripts *entrained*, *cloud*, and *grid* indicate the RH in the entrained, cloudy, and grid-averaged air, respectively. While one can assume that $RH_{cloud}$ is approximately 100 %, $RH_{entrained}$ can vary substantially for a given $RH_{grid}$ if $f$ is not constrained. And when $RH_{entrained}$ is not constrained, the predicted character of entrainment-mixing may be not based in physics. Thus, getting information on $f$ is crucial for the success of the entrainment-mixing parameterization. The authors might want to refer to Jarecka et al. (2009, 2013) on possible pathways to determine $f$. Furthermore, I would like to emphasize that the original data on which the parameterization of this study is based uses mainly an $f = 0.2$ (Luo et al. 2020), while larger $f$ have only been addressed in that study briefly. In fact, Luo et al. (2020) state that different $f$ can change the character of mixing. Without addressing these issues, I cannot support the publication of the manuscript.

**Minor Comments**

L. 1: I would add an "an" before "Entrainment-Mixing".

Ll. 130 –134: Why do you use definition (4) here? Equation (7) might be better as it is directly coupled to the subgrid-scale scheme of the dynamical model. Or do you recommend this approach for models that do not predict $\varepsilon$? Please comment on this decision.

L. 140: The subsaturation is defined as 1-RH, the supersaturation as RH-1, and (5a) requires the supersaturation due to the minus.

Sec. 3: I generally agree with these results. However, the results depend significantly on how well the entrainment-mixing parameterization captures the $f$ dependency. Thus, I do not like to add any more comments at this stage of the publication process.

Ll. 459 – 462: As most models suffer from numerical diffusion, too high supersaturations at the cloud edge are a common problem in most dynamical models, not only large-eddy simulation approaches.

Tab. 3: Add horizontal lines to associate the investigated variables more clearly with the presented values.

Fig. 1: I assume you show the logarithm of the transition scale number here?

**References**

Luo, S., Lu, C., Liu, Y., Bian, J., Gao, W., Li, J., ... & Guo, X. (2020). Parameterizations of entrainment-mixing mechanisms and their effects on cloud droplet spectral width based on numerical simulations. *Journal of Geophysical Research: Atmospheres*, *125*(22), e2020JD032972.

Paluch, I. R. (1979). The entrainment mechanism in Colorado cumuli. *Journal of Atmospheric Sciences*, *36*(12), 2467-2478.

Jarecka, D., Grabowski, W. W., & Pawlowska, H. (2009). Modeling of subgrid-scale mixing in large-eddy simulation of shallow convection. *Journal of the atmospheric sciences*, *66*(7), 2125-2133.

Jarecka, D., Grabowski, W. W., Morrison, H., & Pawlowska, H. (2013). Homogeneity of the subgrid-scale turbulent mixing in large-eddy simulation of shallow convection. *Journal of the atmospheric sciences*, *70*(9), 2751-2767.

---

## Author Comment (AC1)

We would like to thank the reviewer for giving constructive comments/suggestions, which are very helpful in improving the manuscript. We have revised the manuscript based on the comments/suggestions. Below are our detailed responses (blue) to the reviewer's comments/suggestions.

Responses to reviewer #1
Title: Influences of Entrainment-Mixing Parameterization on Numerical Simulations of Cumulus and Stratocumulus Clouds
Authors: Xiaoqi Xu Chunsong, Yangang Liu, Shi Luo, Xin Zhou, Satoshi Endo, Lei Zhu, Yuan Wang

**Summary**

The authors provided a new method for entrainment-mixing parametrization in the LES model. This scheme uses the grid mean RH and can be applied directly in microphysics schemes in the models. They have tested their method in LES version of WRF-Solar simulation for cumulus and stratocumulus clouds. Also, they have conducted the experiments for the sensitivity analysis for different turbulent dissipation rate and aerosol number concentrations. This study provides a good method for parametrization. However, before publication in ACP, the authors have to clarify few questions provided below.

Reply: Thank you very much for your positive evaluation and comments which helps improving the paper.

**Comments**

**Major:** The new scheme is based on the parameters α and the ψ. However, ψ depends on Se (sub saturation of entrained air) which was taken from another model EMPM. My main question is how this new approximated value of Se is validated? It seems that you are taking a parameter from one model and improving another model. It should be validated by some observation or any other reliable source.

In addition, the model results presented in this study should be validated using observation data or a well-established theory.

Reply: There appears some misunderstanding. The Explicit Mixing Parcel Model (EMPM) is only used for developing the entrainment-mixing parameterization. When this parameterization is applied in the Weather Research and Forecasting (WRF) model, we did not use subsaturation of entrained air ($S_e$) from the EMPM model, but use $S_e$ in each grid of the WRF model. Also, in developing the parameterization, we examine a wide range of values for the parameters that affect the entrainment-mixing processes to cover different situations in natural clouds and environment. Therefore, the parameterization is expected to represent different situations (e.g., $S_e$). Furthermore, in the original submission, we used grid-mean relative humidity with a newly developed parameterization (Equation (6)) in Sections 2.1, 3.1-3.4. In the revised manuscript, we add another method, using relative humidity in the entrained air with the parameterization developed by Luo et al. (2020) in Section 3.5.

To address the comments on observational evaluation, we have examined solar irradiance

together with cloud fraction (Figures R1 and R2) to validate the model results. These two figures are added in the revised manuscript (Figures 2 and 4) with detailed discussion.

For the cumulus case (Lines 199-201 and 207-214): "To demonstrate the utility of the model, Figure 2 compares the temporal evolution of the observed and simulated cloud fraction (a) and solar irradiance (b) from the *default* experiment. Considering the difference between the solar irradiances obtained from point measurements and the value representing the simulation domain, the observed solar irradiance at the Southern Great Plains (SGP) Central Facility are compared with the results of central grid point in simulation (Figure 2(b), Figure R1b here). Evidently, although the results of simulation do not fluctuate as much as the observations, the model captures the general behaviours of both cloud fraction and solar irradiance. The general agreement between the simulations and observations lends credence to using the model in further study."

For the stratocumulus case (Lines 247-251): "Figure 4 shows the time series of the domain-averaged cloud fraction and total downward irradiance at the central point in the observation and the *default* experiment from 12:00 UTC to 24:00 UTC (Figure R2 here). Similar to the cumulus case, the simulations compare favourably with the observations, which further reinforces the utility of the LES model."

[Figure]

Figure R1. Time series of (a) domain-averaged cloud fraction and (b) total downward irradiance at the central point from the observation and the *default* experiment in the cumulus case.

[Figure]

Figure R2. Time series of (a) domain-averaged cloud fraction and (b) total downward irradiance at the central point from the observation and the *default* experiment in the stratocumulus case.

---

## Author Comment (AC2)

We would like to thank the reviewer for giving constructive comments/suggestions, which are very helpful in improving the manuscript. We have revised the manuscript based on the comments/suggestions. Below are our detailed responses (blue) to the reviewer's comments/suggestions.

Responses to reviewer #2

Review of "Influences of Entrainment-Mixing Parameterization on Numerical Simulations of Cumulus and Stratocumulus Clouds" by Xu et al. (ACP-2021-937)

The character of entrainment-mixing can have substantial impacts on the microphysical composition of clouds, with commensurate effects on the cloud optical properties and hence their role in the climate system. Nonetheless, most cloud models neglect the natural variability in the mixing process and assume homogeneous mixing as the default. For the application in bulk cloud models, this study develops a parameterization to consider the natural range between homogeneous and extreme inhomogeneous mixing based on simulation data derived from a high-resolution stochastic turbulence model applied in a previous study of the authors (Luo et al. 2020). After development, the parameterization is applied to a cumulus and a stratocumulus case and the results are analyzed. The manuscript is generally well-written, interesting, and of relevance. However, there might be a fundamental flaw in the assumptions made in the parameterization development that can invalidate the entire manuscript, as I will outline further below. Therefore, I cannot recommend the publication of the manuscript at this stage.

Reply: Thank you very much for your comments on our work. Please see the detailed responses to your specific comments below and changes in the revised manuscript. Hope the revised manuscript can eliminate your concern.

**Major Comments**

Developing an entrainment-mixing parameterization that depends on the grid-averaged relative humidity. While I understand that it is necessary to simplify physical processes for parameterization purposes, the presented parameterization glances over crucial factors in the entrainment-mixing process. The entrainment-mixing process depends heavily on the relative humidity RH of the entrained air. While I agree that this variable is not directly accessible in most models, using the grid averaged RH can be misleading if the fraction of entrained air $f$ is unknown. To illustrate this, we make the simplifying assumption that RH mixes linearly. (More rigorous calculations can be based on, e.g., Paluch (1979).) With this simplification, we find that

$$RH_{entrained} = \frac{RH_{grid} - (1-f)RH_{cloud}}{f},$$

where the subscripts *entrained*, *cloud*, and *grid* indicate the RH in the entrained, cloudy, and grid averaged air, respectively. While one can assume that $RH_{cloud}$ is approximately 100%, $RH_{entrained}$ can vary substantially for a given $RH_{grid}$ if $f$ is not constrained. And when $RH_{entrained}$ is not constrained, the predicted character of entrainment-mixing may be not based in physics. Thus, getting information on $f$ is crucial for the success of the entrainment-mixing parameterization. The authors might want to refer to Jarecka et al. (2009, 2013) on possible pathways to determine $f$. Furthermore, I would like to emphasize that the original data on which the parameterization of this study is based

uses mainly an $f = 0.2$ (Luo et al. 2020), while larger $f$ have only been addressed in that study briefly. In fact, Luo et al. (2020) state that different $f$ can change the character of mixing. Without addressing these issues, I cannot support the publication of the manuscript.

Reply: We agree with the reviewer that the entrainment-mixing process depends heavily on the relative humidity of the entrained air, and that the fraction of entrained air ($f$) is a critical parameter.

First, we would like to clarify that the original data on which the parameterization of this study is based uses a range of $f$ values from 0.1 to 0.7, instead of just $f = 0.2$. (Please see Table R1 (Table 1 in Luo et al. (2020)). To avoid such potential misunderstanding, we add some description in Lines 163-165: "It is also worth noting that a wide range of $\varepsilon$, $S_e$, and fraction of entrained air ($f$) are taken into account when establishing the parameterization with EMPM."

Table R1. Table 1 from Luo et al. (2020)

**Table 1**

*Parameters for EMPM Simulations of Entrainment-Mixing Processes*

| Parameter | Value |
| --- | :---: |
| Domain size, $D$ | 20 m |
| Vertical velocity, $w$ | 2 m s$^{-1}$ |
| Entrained blob size, $l$ | 2 m |
| Initial droplet number concentration, $n_d$ | 63.8, 127.6, 191.5, 255.3, 319.2, 383.1 cm$^{-3}$ |
| Entrained air fraction, $f$ | 0.1–0.7 |
| Dissipation rate, $\varepsilon$ | $10^{-5}$, $10^{-4}$, $10^{-3}$, $10^{-2}$, $5 \times 10^{-5}$, $5 \times 10^{-4}$, $5 \times 10^{-3}$, $5 \times 10^{-2}$ m$^2$ s$^{-3}$ |
| Entrainment heights (above cloud base) | 200, 300, 400 m |
| Sounding data (July and August 2014) | 23 days |

Second, we agree that entrained air RH can vary substantially for a given grid-mean RH if $f$ is not constrained and it is a good way to parameterize entrainment-mixing process with the properties of entrained air. Here, the reason for using grid-mean RH is that we want to propose a more convenient choice for most atmospheric models. We agree that it is necessary to verify the results of the entrainment-mixing parameterization shown in the original manuscript by using the entrained air RH with the entrainment-mixing parameterization proposed by Luo et al. (2020)

$$\psi = 107.96 \exp(-0.95 N_L^{-0.35}), \tag{R1}$$

Jarecka et al. (2009) and Jarecka et al. (2013) added an equation to predict $f$ for each grid. In principle, this is a good choice if this method is available in models. Here, to obtain $f$ at 100 m, a

parameterization of $f$ is developed based on the simulations for both the cumulus and stratocumulus cases with a higher resolution of 10 m; the other configurations are the same as those in the experiment *default*. The 10 m-resolution simulation results are then averaged to the resolution of 100 m. Following Xu and Randall (1996), "1 - $f$" can be fitted by the function

$$1 - f = RH_{grid}^{\gamma}\left[1 - \exp\left(-\beta q_c\right)\right],\tag{R2}$$

where $\gamma$ and $\beta$ are empirical parameters. The correlation coefficient between the parameterized and "real" 1 - $f$ is 0.89 with a significant level *p*-value <0.01 (Figure R1).

[Figure]

Figure R1. The fitted 1 - $f$ as a function of the calculated 1 - $f$, where $f$ is fraction of entrained air. The fitted 1 - $f$ is obtained by the fitting function with grid-mean relative humidity ($RH_{grid}$) and cloud water mixing ratio ($q_c$). The black line denotes the 1:1 line.

Then, entrained air RH is calculated following Grabowski (2007) and Jarecka et al. (2009)

$$RH_{entrained} = \frac{RH_{grid} - (1-f)RH_{cloud}}{f},\tag{R3}$$

where the subscripts *entrained*, *cloud*, and *grid* indicate the RH of the entrained, cloudy, and grid point air, respectively. Equations (R1-R3) are applied in the simulations for both the cumulus and stratocumulus cases with different aerosol background (hereafter *new_f* and *new_f_10*).

The above description and discussions are added (Lines 394-421). We also add some discussion on the method used in Jarecka et al. (2009, 2013) in Lines 430-432: "It is worth noting that instead of parameterizing the entrained air fraction, Jarecka et al. (2009) and Jarecka et al. (2013) added an equation to predict $f$ for each grid. In principle, this is a good choice if this method is available in models."

Third, cloud properties in these two extended simulations (*new_f* and *new_f_10*) are compared with the results shown in the original manuscript. Generally, the two different ways of representing entrainment-mixing yield similar cloud microphysical and optical properties (Figures R2-3), with the maximum difference of mean cloud microphysical and optical properties (Table R2) between

using grid RH and entrained air RH being less than 1%. The comparison suggests that the results shown in the original manuscript are reliable. These are discussed in Lines 421-429: "Same as Figures 3 and 5, the temporal evolutions of the cloud physical properties ($q_c$, $N_c$, $r_v$, CWP, and $\tau$) in *default*, *default_10*, *new_f*, and *new_f_10* are shown in Figures 9 and 10. The results are similar to Figures 3 and 5. The mean values of these properties of *new_f* and *new_f_10* for the cumulus and stratocumulus cases are also shown in Table 6, the results of *new* and *new_10* are also shown in the parentheses for the convenience of comparison. The results of *new_f* and *new* are very similar, with the maximum difference being no more than 1%, so are the results of tenfold aerosol background. Such a close agreement suggests that the results of the new entrainment-mixing parametrization with grid-mean RH are reliable."

All the above discussions are added as Section 3.5 in the revised manuscript.

[Figure]

Figure R2. The temporal evolutions of main cloud microphysical and optical properties in all simulation experiments for the cumulus case, including (a) cloud water mixing ratio ($q_c$) (g/kg), (b) cloud droplet number concentration ($N_c$) (/cm$^3$), (c) cloud droplet volume-mean radius ($r_v$) (µm), (d) cloud water path (CWP) (g/m$^2$), and (e) cloud optical depth ($\tau$). In (b), $N_c$ in the experiments *default* and *new_f* are normalized by the maximum cloud droplet concentration ($N_{cmax}$) from *default*, respectively; $N_c$ in the experiments *default_10* and *new__f_10* are normalized by $N_{cmax}$ from *default_10*, respectively. *new_f* and *new_f_10* are the experiments using entrained air relative

humidity.

[Figure]

Figure R3. The temporal evolutions of main cloud microphysical and optical properties in all simulation experiments for the stratocumulus case, including (a) cloud water mixing ratio ($q_c$) (g/kg), (b) cloud droplet number concentration ($N_c$) (/cm$^3$), (c) cloud droplet volume-mean radius ($r_v$) (μm), (d) cloud water path (CWP) (g/m$^2$), and (e) cloud optical depth ($\tau$). In (b), $N_c$ in the experiments *default* and *new* are normalized by the maximum cloud droplet number concentration ($N_{cmax}$) from *default*, respectively; $N_c$ in the experiments *default_10* and *new_10* are normalized by $N_{cmax}$ from *default_10*, respectively. *new_f* and *new_f_10* are the experiments using entrained air relative humidity.

Table R2. Cloud water mixing ratio ($q_c$), cloud droplet number concentration ($N_c$), cloud droplet volume-mean radius ($r_v$), cloud water path (CWP), cloud optical depth ($\tau$) the cumulus (Cu) and stratocumulus (Sc) cases. *new_f* and *new_f_10* are the simulations using entrained air RH with Equation (R1) while *new* and *new_10* are the simulations using grid-mean RH as shown in the original manuscript.

| | Cu | | Sc | |
|---|---|---|---|---|
| | *new_f* (*new*) | *new_f_10* (*new_10*) | *new_f* (*new*) | *new_f_10* (*new_10*) |
| $q_c$(g/kg) | 0.44 (0.44) | 0.57 (0.57) | 0.11 (0.11) | 0.13 (0.13) |
| $N_c$(cm$^{-3}$) | 35.52 (35.53) | 270.56 (271.16) | 28.08 (28.12) | 202.99 (203.50) |
| $r_v$(μm) | 13.30 (13.29) | 7.10 (7.09) | 9.60 (9.57) | 5.21 (5.22) |
| CWP(g/m$^2$) | 143.15 (144.25) | 185.95 (187.13) | 30.16 (29.92) | 43.32 (43.13) |
| $\tau$ | 13.00 (13.02) | 31.08 (31.11) | 3.89 (3.89) | 9.93 (9.96) |

**References**

Grabowski, W. W.: Representation of turbulent mixing and buoyancy reversal in bulk cloud models, Journal of the atmospheric sciences, 64, 3666-3680, 2007.

Jarecka, D., Grabowski, W. W., and Pawlowska, H.: Modeling of Subgrid-Scale Mixing in Large-Eddy Simulation of Shallow Convection, Journal of the Atmospheric Sciences, 66, 2125-2133, 10.1175/2009jas2929.1, 2009.

Jarecka, D., Grabowski, W. W., Morrison, H., and Pawlowska, H.: Homogeneity of the subgrid-scale turbulent mixing in large-eddy simulation of shallow convection, J. Atmos. Sci., 70, 2751-2767, 2013.

Luo, S., Lu, C., Liu, Y., Bian, J., Gao, W., Li, J., Xu, X., Gao, S., Yang, S., and Guo, X.: Parameterizations of Entrainment-Mixing Mechanisms and Their Effects on Cloud Droplet Spectral Width Based on Numerical Simulations, Journal of Geophysical Research: Atmospheres, 125, e2020JD032972, 2020.

Xu, K.-M. and Randall, D. A.: A semiempirical cloudiness parameterization for use in climate models, Journal of the atmospheric sciences, 53, 3084-3102, 1996.

**Minor Comments**

L. 1: I would add an "an" before "Entrainment-Mixing".

Reply: Thanks, and added.

Ll. 130 –134: Why do you use definition (4) here? Equation (7) might be better as it is directly coupled to the subgrid-scale scheme of the dynamical model. Or do you recommend this approach for models that do not predict $\varepsilon$ ? Please comment on this decision.

Reply: Thank you for your comment. We agree that Equation (7) is better to be directly coupled to the subgrid-scale scheme. We remove the results with Equation (4) in the original manuscript and replace these with the results with Equation (7). The general conclusions do not change (Section 3.1-3.4).

L. 140: The subsaturation is defined as 1-RH, the supersaturation as RH-1, and (5a) requires the supersaturation due to the minus.

Reply: We have revised the phrases accordingly.

Sec. 3: I generally agree with these results. However, the results depend significantly on how well the entrainment-mixing parameterization captures the $f$ dependency. Thus, I do not like to add any more comments at this stage of the publication process.

Reply: The results of the parameterization with grid-mean RH in the original manuscript are verified by the parameterization with entrained air RH in Section 3.5 in the revised manuscript. Please also see the reply to Major comment.

Ll. 459 – 462: As most models suffer from numerical diffusion, too high supersaturations at the cloud edge are a common problem in most dynamical models, not only large-eddy simulation approaches.

Reply: We have changed the sentence to "Note that the new entrainment-mixing parameterization could be more important in the models if the relative humidity near the cloud is more accurately simulated, because numerical diffusion may spuriously humidify the entrained air".

Tab. 3: Add horizontal lines to associate the investigated variables more clearly with the presented values.

Reply: Added.

Fig. 1: I assume you show the logarithm of the transition scale number here?

Reply: Yes. Sorry for that and we have replotted Figure 1 in the revised manuscript.

---

## Referee Report (RR1)

**Review of "Influences of an Entrainment-Mixing Parameterization on Numerical Simulations of Cumulus and Stratocumulus Clouds" by Xu et al. (ACP-2021-937)**

While I am generally pleased with the addition of the entrainment fraction parameterization and the additional analyses, I still have some major concerns, as outlined below. Thus, I would like the authors to address the following concerns below before I can recommend the publication of this manuscript.

**Major Comments**

*Is the parameterized entrainment fraction suitable for microscale processes?* Of course, the fitting seems to be successful (Fig. 8), and I like the idea of parameterizing the entrainment fraction based on the grid-scale relative humidity and cloud water mixing ratio. However, I doubt that this parameterization is suitable for a microscale process, where local shear and buoyancy drive turbulence generation and entrainment. In fact, Xu and Randall (1996) developed the applied parameterization for climate models in the 1990s, i.e., for representing entire subgrid-scale clouds at a resolution of several tens to hundreds of kilometers, while the authors apply it for subgrid-scale processes below 100 m. Finally, I wonder why the data on the x-axis of Fig. 8 is not evenly spaced? The calculated $(1-f)$ values should have values between 0 and 1 with a spacing of 0.01, which should be visible in the plot. Or is there some post-processing not mentioned in the manuscript?

*Extension of the parameterization to account for entrainment fraction.* In Luo et al. (2020), the authors showed that the entrainment fraction impacts the subsequent mixing process. Why is the entrainment fraction not considered in their parameterization (6)?

**References**

Luo, S., Lu, C., Liu, Y., Bian, J., Gao, W., Li, J., ... & Guo, X. (2020). Parameterizations of entrainment-mixing mechanisms and their effects on cloud droplet spectral width based on numerical simulations. *Journal of Geophysical Research: Atmospheres*, *125*(22), e2020JD032972.

Xu, K. M., & Randall, D. A. (1996). A semiempirical cloudiness parameterization for use in climate models. *Journal of the atmospheric sciences*, *53*(21), 3084-3102.

---

## Author Response (AR2)

**We would like to thank the reviewer for giving constructive comments/suggestions, which are very helpful in improving the manuscript. We have revised the manuscript based on the comments/suggestions. Below are our detailed responses (blue) to the reviewer's comments/suggestions.**

Responses to reviewer

Review of "Influences of Entrainment-Mixing Parameterization on Numerical Simulations of Cumulus and Stratocumulus Clouds" by Xu et al. (ACP-2021-937)

While I am generally pleased with the addition of the entrainment fraction parameterization and the additional analyses, I still have some major concerns, as outlined below. Thus, I would like the authors to address the following concerns below before I can recommend the publication of this manuscript.

Reply: Thank you very much for your comments on our work. Please see the detailed responses to your specific comments below and the changes in the revised manuscript. Hope the revised manuscript can eliminate your concern.

**Major Comments**

*Is the parameterized entrainment fraction suitable for microscale processes?* Of course, the fitting seems to be successful (Fig. 8), and I like the idea of parameterizing the entrainment fraction based on the grid-scale relative humidity and cloud water mixing ratio. However, I doubt that this parameterization is suitable for a microscale process, where local shear and buoyancy drive turbulence generation and entrainment. In fact, Xu and Randall (1996) developed the applied parameterization for climate models in the 1990s, i.e., for representing entire subgrid-scale clouds at a resolution of several tens to hundreds of kilometers, while the authors apply it for subgrid-scale processes below 100 m. Finally, I wonder why the data on the x-axis of Fig. 8 is not evenly spaced? The calculated ( 1 – f ) values should have values between 0 and 1 with a spacing of 0.01, which should be visible in the plot. Or is there some post-processing not mentioned in the manuscript?

Reply: We agree with the reviewer that local shear ($dw/dz$) and buoyancy ($B$) may drive turbulence generation and entrainment for a microscale process. Therefore, $dw/dz$ and $B$ are used to fit "1 - $f$" (Figure R1). Since some values of $dw/dz$ and $B$ are negative, their absolute values are

taken in the fitting with the power law function. Figure R1(a) shows that the parameterization with $dw/dz$ and $B$ has poor performance with correlation coefficient ($R$) of 0.17 and root-mean-square error (RMSE) of 0.20. Figure R1 also shows the parameterization relating "1 - $f$" to relative humidity ($\text{RH}_{\text{grid}}$) and cloud water mixing ratio ($q_c$), which is used in our simulation; $R$ and RMSE are 0.89 and 0.10, respectively. Therefore, this parameterization is much better than that based on $dw/dz$ and $B$.

To further confirm that the parameterization using $\text{RH}_{\text{grid}}$ and $q_c$ is a good choice, Figure R1(b) furthers shows the results by adding $dw/dz$ and $B$ to the parameterization using $\text{RH}_{\text{grid}}$ and $q_c$. It is interesting to find that the addition of $dw/dz$ and $B$ neither increases $R$ nor decreases RMSE. Therefore, using $\text{RH}_{\text{grid}}$ and $q_c$ to parametrize "$1 - f$" is good and reasonable, at least for the cases in this study. The corresponding discussions are added in Lines 418-423: "Considering that local shear ($dw/dz$) and buoyancy ($B$) may drive turbulence generation and entrainment for a microscale process, the two quantities are also used to fit "1 - $f$" except for $\text{RH}_{\text{grid}}$ and $q_c$. However, the addition of $dw/dz$ and $B$ to Equation (11) does not increase $R$. Therefore, using $\text{RH}_{\text{grid}}$ and $q_c$ to parametrize "$1 - f$" is good and reasonable for a microscale process."

By the way, we recognize the possible scale mismatch concern and the Xu and Randall (1996) developed their parameterization for climate models. That's why our parameterization of "$1 - f$" is developed based on the large eddy simulations (LES) with a higher resolution of 10 m, instead of directly using the Xu-Randall parameterization.

In Fig. 8, the original data of "$1 - f$" are used without binning the data with a spacing of 0.01. The non-uniform distribution of the data points is because the occurrence frequency of individual mixing fractions is not the same. For example, no original data is in the range of 0.9 - 1 and over half of the data is smaller than 0.3.

[Figure]

Figure R1. The fitted $1 - f$ as a function of the calculated $1 - f$. The fitted $1 - f$ is obtained by the fitting functions with different combinations of grid-mean relative humidity ($RH_{grid}$), cloud water mixing ratio ($q_c$), buoyancy ($B$), and vertical wind shear ($dw/dz$). The black lines denote the 1:1 line. Each legend provides the correlation coefficient ($R$), the fitting function, and the root-mean-square error (RMSE). All the $p$ values are smaller than 0.01.

*Extension of the parameterization to account for entrainment fraction.* In Luo et al. (2020), the authors showed that the entrainment fraction impacts the subsequent mixing process. Why is the entrainment fraction not considered in their parameterization (6)?

Reply: We agree with the reviewer that the entrainment fraction is important for entrainment-mixing processes; however, it is very hard to obtain the accurate entrainment faction in most numerical models, including large eddy simulation (LES) models, and the parameterization proposed in Luo et al. (2020) was established for these models. To the authors' knowledge, only in the LES model used by Jarecka et al. (2009) and Jarecka et al. (2013), entrainment faction was explicitly available by adding an equation to predict entrainment fraction for each grid. Since entrainment faction is hard to be determined in other models and in observations, Lehmann et al. (2009) defined transition length for Damköhler Number equal to 1. They argued that "The transition length scale separates the inertial subrange into a range of length scales for which mixing between ambient dry and cloudy air is inhomogeneous, and a range for which the mixing is homogeneous." We understand that the transition length is not perfect, but should be a good choice when entrainment fraction is not available. Therefore, Luo et al. (2020) developed a parameterization based on transition scale defined as the ratio of the transition length to the Kolmogorov scale (Kumar et al., 2013; Lu et al., 2011). Entrainment fraction was considered implicitly in Luo et al. (2020)'s parameterization, because the Explicit Mixing Parcel Model simulations with different entrainment fraction were used to develop this parameterization.

Again, it is better to explicitly consider entrainment fraction as in Jarecka et al. (2009) and Jarecka et al. (2013) in principle. The method used in our study is an alternative way to represent entrainment-mixing process when the prognostic entrainment fraction is not available.

We have added some discussions in the revised manuscript (Lines 435-438).

References

Lehmann, K., Siebert, H., and Shaw, R. A.: Homogeneous and inhomogeneous mixing in cumulus clouds: dependence on local turbulence structure, J. Atmos. Sci., 66, 3641-3659, doi:10.1175/2009JAS3012.1, 2009.

Jarecka, D., Grabowski, W. W., and Pawlowska, H.: Modeling of Subgrid-Scale Mixing in Large-Eddy Simulation of Shallow Convection, Journal of the Atmospheric Sciences, 66, 2125-2133,

10.1175/2009jas2929.1, 2009.

Jarecka, D., Grabowski, W. W., Morrison, H., and Pawlowska, H.: Homogeneity of the subgrid-scale turbulent mixing in large-eddy simulation of shallow convection, J. Atmos. Sci., 70, 2751-2767, 2013.

Kumar, B., J. Schumacher, and R. Shaw (2013), Cloud microphysical effects of turbulent mixing and entrainment, Theor. Comp. Fluid Dyn., 27, 361-376.

Lu, C., Liu, Y., and Niu, S.: Examination of turbulent entrainment-mixing mechanisms using a combined approach, J. Geophys. Res., 116, D20207, doi:10.1029/2011JD015944, 2011.